# *AuthAR*: Concurrent Authoring of Tutorials for AR Assembly Guidance

Matt Whitlock[*]
Autodesk Reasearch, Toronto
University of Colorado Boulder

George Fitzmaurice[†]
Autodesk Research, Toronto

Tovi Grossman [‡]
Autodesk Research, Toronto
University of Toronto

Justin Matejka [§]
Autodesk Research, Toronto

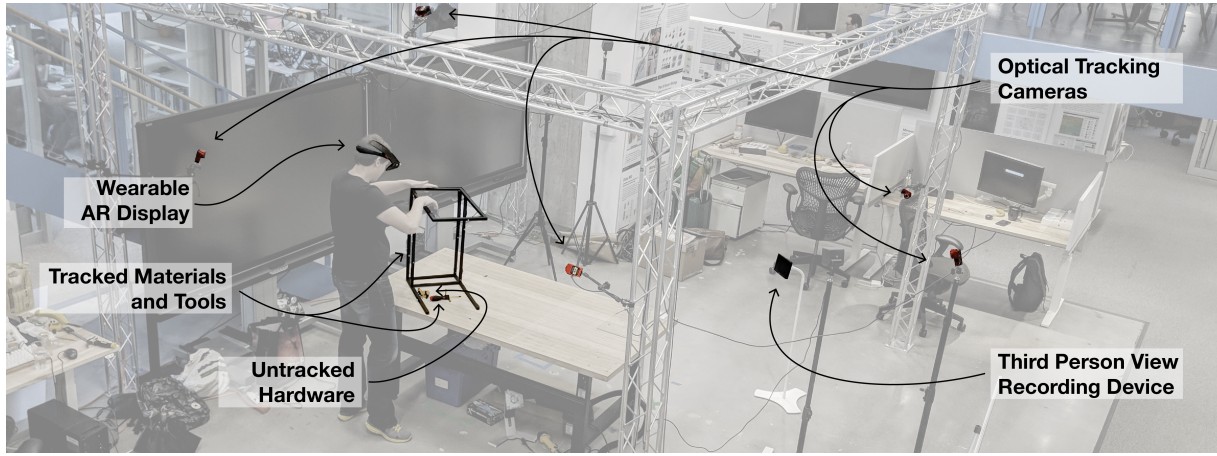

Figure 1: Overview of the *AuthAR* system setup, highlighting the key hardware components.

## ABSTRACT

Augmented Reality (AR) can assist with physical tasks such as object assembly through the use of situated instructions. These instructions can be in the form of videos, pictures, text or guiding animations, where the most helpful media among these is highly dependent on both the user and the nature of the task. Our work supports the authoring of AR tutorials for assembly tasks with little overhead beyond simply performing the task itself. The presented system, *AuthAR* reduces the time and effort required to build inter-active AR tutorials by automatically generating key components of the AR tutorial while the author is assembling the physical pieces. Further, the system guides authors through the process of adding videos, pictures, text and animations to the tutorial. This concurrent assembly and tutorial generation approach allows for authoring of portable tutorials that fit the preferences of different end users.

**Keywords:** Augmented reality, content authoring, assembly tutorials, gaze input, voice input

**Index Terms:** H.5.m [Information Interfaces and Presentation]: Miscellaneous

## 1 INTRODUCTION

Physical task guidance can be delivered via Augmented Reality (AR) since assembly often requires both hands and continuous attention to the task. Additionally, assembly tutorials have instructions directly associated with physical objects, so AR can reduce the need for excessive context switching between the instructions and the physical structure by projecting those instructions into the environment. These benefits have been demonstrated in fields such as Facilities

---
[*]e-mail: matthew.whitlock@colorado.edu
[†]e-mail: george.fitzmaurice@autodesk.com
[‡]e-mail: tovi@dgp.toronto.edu
[§]e-mail: justin.matejka@autodesk.com

Management [20], Maintenance [44], and Internet of Things (IoT) device management [14, 21]. Additionally, prior work in AR assembly guidance has shown that these benefits can translate to carrying out assembly tasks [2, 18, 21, 38].

While significant previous work has looked at the benefits of following tutorials in AR, much less has looked at how to author these tutorials. Beyond the technical requirements of an authoring interface, an ideal tutorial may look different depending on the end user of the tutorial. This problem is exacerbated in AR as there are many different modalities in which tutorial content can be presented. While one person may appreciate guiding animations in AR, another may prefer static text and images, and yet another may prefer video tutorials from one or multiple perspectives.

With *AuthAR*, we present a system for building tutorials for assembly tasks that can accommodate the needs of these different types of end users. *AuthAR* generates video, and pictorial representations semi-automatically while the tutorial author completes the task. Furthermore, *AuthAR* allows tutorial authors to create and refine a tutorial *in situ*, integrating content authoring into the process of completing the task. This approach adds little additional overhead and reduces the need for post-processing of the tutorial.

This paper presents the *AuthAR* system for generating mixed media assembly tutorials. Informed by prior work on content/tutorial authoring, and tutorial playback and walkthrough, we build the system with an eye toward non-obtrusive content authoring and generation of important components for tutorial playback, summarized in a set of design guidelines. We validate the system's ability to create a tutorial by stepping through the process of creating a tutorial to build a laptop stand, automatically generating an XML representation of the tutorial. Initial observations suggest the tool will be valuable, and possible ways the system could be extended and refined in future iterations.

## 2 RELATED WORK

*AuthAR* builds on prior research in the areas of AR tutorials and content authoring, as well as principles of mixed media tutorial design.

## 2.1 AR Tutorials

AR is often applied to assembly tasks for its ability to project instructions into the environment such that they are spatially relevant [19, 22]. Syberfeldt et al. demonstrate assembly of a 3D puzzle and provide evidence that AR supports faster assembly than traditional methods [39]. Similarly, Henderson et al. demonstrate the use of projected guidance for engine assembly [18]. Prior work on AR guidance design has shown that abstract representations (3D text, arrows) can be more effective for complex tasks than a more concrete representation (virtual models of the pieces and fasteners), which is sensible for simpler tasks [34]. In that work, the authors found information-rich 2D representations to be more effective than either of the AR representations in some cases.

One theory is that AR is only justified when the task is sufficiently difficult, such that the time to process the information is insignificant compared to the time to perform the task [33]. So even for physical tasks in which instructions' spatial relevance could be increased by projecting into the environment, tutorials should provide users the ability to view the step(s) in a more familiar picture/video format when needed or preferred. The need for these mixed media tutorials is apparent, however little work has explored *authoring* of these tutorials for physical tasks.

## 2.2 AR Content Authoring

Outside of the realm of tutorial authoring, numerous systems have explored content creation in augmented reality to abstract low-level programming from the creator, lowering the threshold for participation with AR. Many AR content authoring systems give users a collection of 3D models to place and manipulate in an environment or to overlay on a video stream, allowing users to create AR content without programming expertise generally required to build such scenes [4, 11, 12, 24, 26, 30]. Other AR content creation systems target specific end users for participation by domain experts in areas such as museum exhibition curation [36] tour guidance [3, 27], and assembly/maintenance [33, 41]. Roberto et al. provides a survey of existing AR content creation tools, classifying tools by standalone nature and platform-dependence [35].

Of particular interest to us are authoring tools that enable creation of training experiences. Built on Amire's component-based framework [12], Zauner et al. presents a tool to enable authoring of assembly task guides in augmented reality [43]. With this tool, the author puts visually tracked pieces together hierarchically to create an assembly workflow assistant in AR. Alternatively, the expert can collaborate remotely, rather than creating a training experience ahead of time [41]. In this scenario, the expert can annotate the live video feed provided by the trainee's AR headset for varying levels of guidance on-the-fly. These systems require explicit enumeration of every component to be added to the created scene whereas our system generates content semi-automatically (segmenting video, recording changes to transform and detecting use of a tool) where possible. Moreover, our system only requires this manual input for augmentation and refinement of the tutorial while the bulk of authoring is done *in situ*.

## 2.3 Mixed Media Tutorials

Within the domain of software tutorials, Chi et al. provides design guidelines for mixed media tutorial authoring with their MixT system for software tutorials [7]. They list scannable steps, legible videos, visualized mouse movement and giving control to the user on which format to view as important components of a mixed media tutorial. Carter et al. echos this sentiment, with their ShowHow system for building a tutorial of videos and pictures taken from an HMD, noting that mixing different media is important in lieu of relying on a single media type [6]. Our work builds upon these concepts of mixed media tutorials but applies them to AR authoring of physical task tutorials. The rest of this subsection discusses the use of three popular media used for both software and physical task tutorials: videos, images, and interactive guidance.

### 2.3.1 Video

Prior work on video-based tutorials has applied different strategies to the challenges of video segmentation and multiple perspectives. DemoCut allows for semi-automatic video segmentation such that these demonstration videos are appropriately concise without requiring significant post-processing [8]. Chronicle allows for video tutorials based on the working history of a file [16]. As the file changes, the Chronicle system generates a video tutorial of how the file changed. For physical tasks, Nakae et al. propose use of multiple video perspectives (1st person, 3rd person, overhead) to record fabrication demonstrations and semi-automatic generation of these videos [29].

### 2.3.2 Images

Prior work also uses captioned and augmented images for physical task and software tutorials. Image-based tutorials can be generated automatically from demonstration as is done with TutorialPlan, a system that builds tutorials to help novices learn AutoCAD [25]. Images can also represent groups of significant manipulations (such as multiple changes to the saturation parameter) as Grabler et al. demonstrated for GIMP tutorials [15]. For physical tasks, prior work explored use of AR to retarget 2D technical documentation onto the object itself, providing spatially relevant augmentations [28].

### 2.3.3 Interactive Overlays

Interactive tutorials guiding users where to click, place objects, and even move their hands have become increasingly popular. For example, EverTutor automatically generates tutorials for smart phone tasks such as setting a repeating alarm or changing font size based on touch events [40]. They found improved performance and preference toward these tutorials over text, video, and image tutorials. In the realm of physical assembly tutorials, use of visual and/or depth tracking allows for automatic generation of tutorials based on changes to location and rotation [5, 10, 13, 39]. Further, interactive tutorial authoring can include tracking of hand position [32], and project green and red onto end users' hands to indicate correct and incorrect positioning respectively [31]. DuploTrack allows users to author a tutorial for creating Duplo block models using depth sensing to infer positions and rotations automatically, and projective guidance is available to end users of the tutorial [17].

## 3 DESIGN GUIDELINES

Our design of the *AuthAR* system was grounded by our exploration of the assembly task design space, and a study of related research and investigation into the difficulties associated with the process of generating tutorials (both for "traditional" media, as well as AR). Below we describe the design guidelines we followed when making decisions about the implementation of our system.

**D1: Non-Intrusive/Hand-Free.** It is important that the author and assembler be able to perform the assembly task without being burdened by the tutorial apparatus or interface. Though many AR/VR interfaces are either mediated by a handheld device or require use of freehand gestures for input, assembly tasks often require the use of both hands in parallel. For this reason, we prioritize hands-free interaction with the system such that users can always keep their hands free for assembly.

**D2: Multiple Representations.** Prior studies [6, 40] have shown that different representations (text, static pictures, video, animations, etc.) can all be valuable for people following along with a tutorial. Our system should allow authors to document their tutorial using multiple media types to best capture the necessary information.

**D3: Adaptive Effort.** Manual content creation in AR allows for high expressivity but is time-consuming and can add complexity.

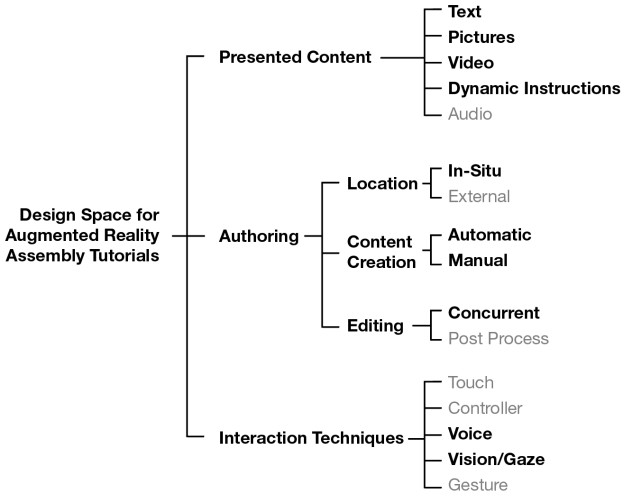

Figure 2: Design Space for AR Assembly tutorials. The design decisions we made are highlighted in black.

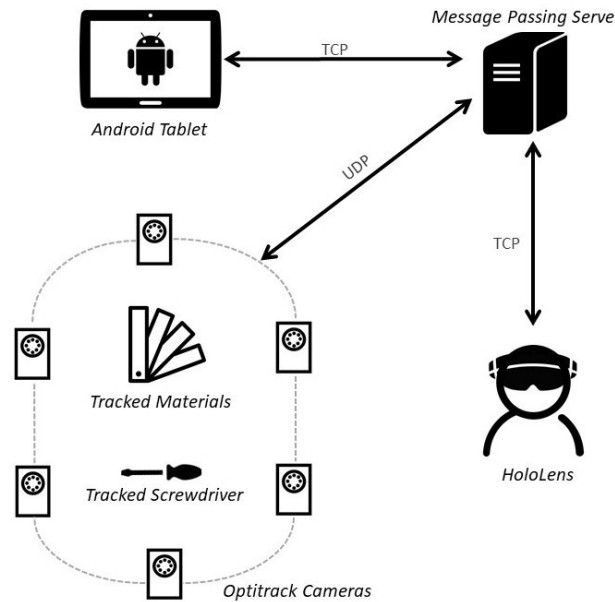

Figure 3: *AuthAR* System Diagram. A message passing server sends position data of materials and a screwdriver from coordinated Optitrack cameras to the HoloLens. The HoloLens sends video segmenting commands to the Android tablet through the server.

Automatic content creation tools are easier to use, however have the side effect of limiting the author's creative control. Our tool should let authors move between automatic and manual creation modes to get the benefits of both models. In the most "automatic" case, an author should be able to generate a tutorial by doing little more than simply completing the assembly task as they would normally.

**D4: Real Time and *In Situ* Authoring.** With the ability to generate, refine and augment tutorial step representations, our tool should allow authors to create the tutorial while performing the task and make necessary tweaks directly after each step. This form of contextual and *in situ* editing allows authors to add callouts or make changes while they are easy to remember and reduces the need for post-processing of tutorials at a desktop computer.

## 4 DESIGN SPACE OF AR ASSEMBLY TASK TUTORIALS

The design space of tutorials for assembly tasks in augmented reality is broad, with many dimensions of variability both in how the instructions are presented to an end-user and how they are authored. To inform the design of our system and explore how to best achieve our design goals, we first mapped out some areas of this design space most relevant to our work (Figure 2).

*Presented Content*

Perhaps the most important dimension of variability in the AR-based assembly task tutorial design space is how the tutorial information is presented within the AR environment. Like more "traditional" tutorial delivery mediums, the AR environment is able to present static text and images, as well as show explanatory videos. A most straightforward (and still useful) application of AR technology for sharing assembly tutorials would be to display relevant information about the task as a heads-up display (HUD) in the user's headset, leaving their hand's free to perform the task. Unique to AR however is the ability to spatially associate these "traditional" elements with points or objects in the physical space. Further, an AR tutorial can present the assembly instructions "live", by displaying assembly guidance graphics or animations interwoven into the physical space. With our system we chose to use traditional text, pictures, and video presentation methods in addition to dynamic instructions, since they each have their unique benefits **(D2)**.

In order for the HoloLens to listen for the keyword "Stop Recording" during first/third person video recording, it cannot simultaneously record dictation to complement the video. For this reason,

the current form of AuthAR records muted videos, but future iterations with additional microphone input capability would rectify this. With use of audio in content authoring infeasible, text serves to supplement muted video.

*Authoring Location*

When it comes to authoring the tutorial, the content could be constructed in situ, that is, in the same physical space as the task is being performed, or in an external secondary location such as a desktop computer. Our work explores in situ authoring to reduce the required effort **(D3)** and maintain context **(D4)**.

*Content Creation*

The content for the tutorial can be automatically captured as the authors moves through the assembly steps, or manually created by explicitly specifying what should be happening at each step. Automatically generating the tutorials streamlines the authoring and content creation process, but limits expressivity of the tutorial instructions. We automatically capture as much as possible, while allowing manual additions where the author thinks they would be helpful **(D3)**.

*Content Editing*

Another aspect of tutorial authoring is how the content is edited. A traditional practice is to capture the necessary content first, then go back and edit for time and clarity in post-processing. Alternately, the content can be edited concurrently with the process of content collection–which if implemented poorly could negatively impact the flow of the creation process. In the best case, this results in having a completed tutorial ready to share as soon as the author has completed the task themselves. We focus on creating a well-designed concurrent editing process **(D1)**.

*Interaction Techniques*

Interacting with the tutorial system, both in the creation of the tutorial as well as when following along, can be accomplished through many means. AR applications can use touch controls, gestures, or

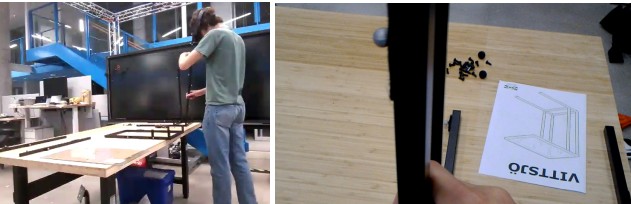

Figure 5: Simultaneous 3rd person video recording from the Android tablet (left) and 1st person video recording from the HoloLens (right).

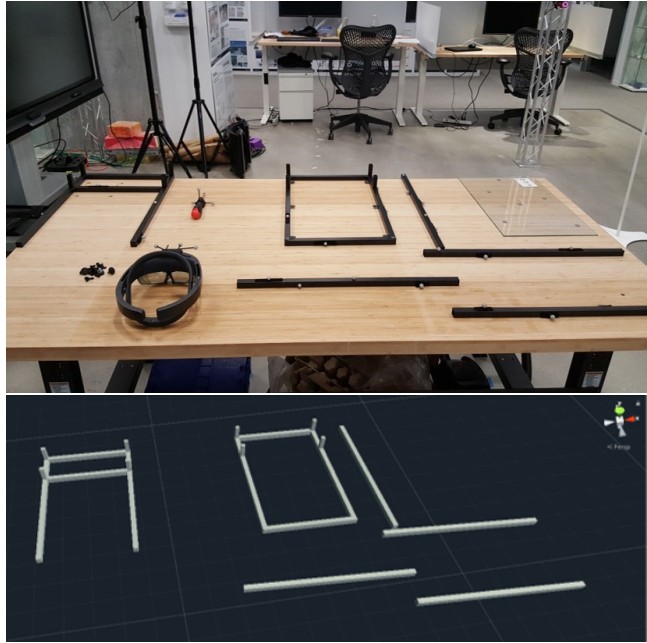

Figure 4: Example configuration of tracked materials in the physical environment (top) and transform data streaming from Optitrack to the Message Passing Server to provide positional tracking of invisible renderings (bottom). Within *AuthAR*, the virtual components are overlaid on the physical pieces, such that the physical components are interactive.

dedicated hardware controllers. However, for assembly tasks it is desirable to have both hands available at all times for the assembly task – rather than have them occupied by control of the tutorial authoring system (**D1**). For that reason, we exclusively use voice and gaze controls.

## 5 THE AUTHAR SYSTEM

*AuthAR* is a suite of software tools that allow tutorial authors to generate AR content (Figure 1). Optitrack motion capture cameras visually track marked assembly materials and a screwdriver, adding changes to position, transform and locations of screws added with a tracked screwdriver to the tutorial automatically. The HoloLens captures first person videos and images and a mounted Android tablet simultaneously captures third person video. The HoloLens also guides authors through the process of creating the tutorial and allows for gaze- and voice-based interaction to add and update content. This includes addition of augmentations such as location-specific callout points, marking locations of untrackable screws, and specification of images as "negative examples" (steps that should be avoided) or warnings.

### 5.1 System Architecture

*AuthAR's* system architecture consists of three components—the Microsoft HoloLens, a Samsung Tab A 10.1 Android tablet and a server running on a desktop computer all on the same network (Figure 3). As a proxy to sufficient object recognition and point-cloud generation directly from the headset, we use relatively small (approximately 1cm) Optitrack visual markers for detection of material position and rotation. In developing *AuthAR*, we envisioned headsets of the future having onboard object recognition [9, 24]. As a proxy to such capabilities, we implement a networked system to coordinate object positions generated by Optitrack's Motive software and the HoloLens to make the physical objects interactive. To provide this

interactivity, the HoloLens registers virtual replicas of each piece and overlays these models as invisible renderings at the position and rotation of the tracked object. This invisible object takes the same shape as the physical object but has its visual rendering component disabled. It engages the HoloLens raycast and gives the user the illusion of placing virtual augmentations on physical objects. Additionally, we add a tracked handle to a screwdriver for tracking to infer screw events.

The server connects to the Optitrack host via UDP and continues to update object and tool transforms. This server sends the data to the HoloLens via 64-character messages and the HoloLens' representation of the transform is updated accordingly. The Android tablet simply serves to record 3rd person video of the tutorial. When the HoloLens starts and stops recording 1st person demonstration, a message is passed to the server and then to the Android tablet to toggle recording. Throughout the paper, we demonstrate usage of *AuthAR* to build a tutorial for assembly of an Ikea laptop stand[1]. Parts have been outfitted with Optitrack visual markers and defined as rigid bodies in Optitrack's Motive software. Simple representations of these parts are by the HoloLens as invisible colliders, so the physical components act as interactive objects in AR (Figure 4).

Though this configuration is specific to the laptop stand, this approach is easily extensible to other assembly workflows. Users simply define combinations of visual markers as rigid bodies and build simplified models of the individual parts. In this case, these models are combinations of cubes that have been scaled along X, Y and Z axes, giving rough approximations of the parts' shapes. This initial step of predefining the shapes of these parts allows fully *in situ* editing.

### 5.2 Interaction Paradigm

To avoid encumbering the assembly process with tutorial generation steps, interaction with *AuthAR* involves only voice, gaze, and use of materials and tools. This allows the user to always keep their hands free to build. By using only voice- and gaze-based interaction, we also eliminate the need for an occluding visual interface of menus and buttons, avoiding interference with the physical tutorial tasks. For flexibility, the user can add augmentations at any point while building the tutorial. However, to encourage faster onboarding, we guide the user through a two-phase process for each step: *Step Recording* and *Step Review*.

To provide this guidance, we implemented a three-part Heads-Up Display (HUD). The top-right always displays both the current state and the command to advance to the next stage, the top-left shows available commands within *Step Review* mode, and the middle is reserved for notifications and prompts for dictation. A quick green flash indicates that the user has successfully moved to the next state.

*Step Recording: Automatically Generated Features*

---

[1]https://www.ikea.com/us/en/p/vittsjoe-laptop-stand-black-brown-glass-00250249/

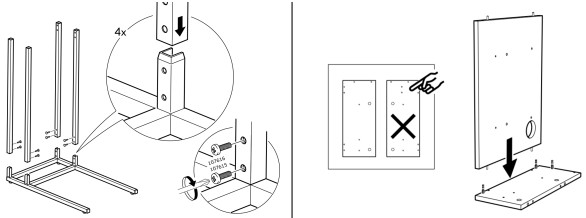

Figure 6: Example usage of callout points in paper-based instructions. Callout points draw attention to the alignment of the holes on the materials (left). Instructions can convey negative examples of incorrect object orientation (right). Images from the assembly instructions for an Ikea laptop stand.

When the user says "Start Recording" *AuthAR* begins recording changes to object transforms such that moving the physical objects maps directly to manipulating the virtual representations of those objects in the tutorial. This command also initiates video recording using the HoloLens' built-in camera and the tablet's $1^{st}$ person perspective (Figure 5). *AuthAR* also records when the screwdriver's tip comes in contact with an object piece and generates a screw hole on that object (though not displayed until the Review Mode). To do this we add a tracked attachment to the handle, similar to what was done for Dodecapen [42] and SymbiosisSketch [1] to provide the position of the tip based on the orientation of the attachment.

Given real-time streaming of position data of the screwdriver's handle and *a priori* knowledge of the length from the handle to the tip as 10.5 cm, we calculate the position of the screwdriver's tip as 10.5 cm forward from the handle. When the user says "Finish Recording", the HoloLens prompts the user for a step description and records dictation. When description is complete, the user enters an idle state until they are ready for review.

*Step Review: Manually Added Features*

After a tutorial author has completed a step recording, they can they can say "*Review Step*" to enter review mode for the step. The $1^{st}$ person video just recorded plays on a loop across from the author, automatically repositioning itself such that the author can look up at any point and see the video directly in front of them. This allows the author to draw upon their own experience when adding manual augmentations. Existing augmentations (e.g., callout points and fasteners) shift into focus by getting larger or expanding when the user is looking closer to that augmentation than any other—this eliminates the need to focus on small points (approximately 3cm) to engage with them. When looking toward a particular object, the available commands to update the augmentation are shown in the top-right of the Heads-up Display.

After recording the step, the tutorial author may want to draw the tutorial user's attention to a particular point, similar to how this is done on 2D paper-based instructions (Figure 6). To do this, the author focuses the gaze-based cursor on the specific point on the tracked object where the callout point should go and says "*Add Point*". This adds a small virtual sphere anchored by its relative position on the object. The process of adding a captioned image (Figure 7), begins when the user looks toward a callout point and says "*Add Picture*". The system then starts a countdown from three in the middle of the heads-up display to indicate that a picture is about to be taken, and then displays "Hold still!" when the countdown is complete, at which point the HoloLens captures the current frame.

The collected image is saved and immediately loaded over the callout point. The author can then say "*Add Text*", and a prompt in the middle of the heads-up display shows "Speak the image caption" and the HoloLens begins listening for recorded dictation. After a pause of 3 seconds, the HoloLens finishes recording and immediately

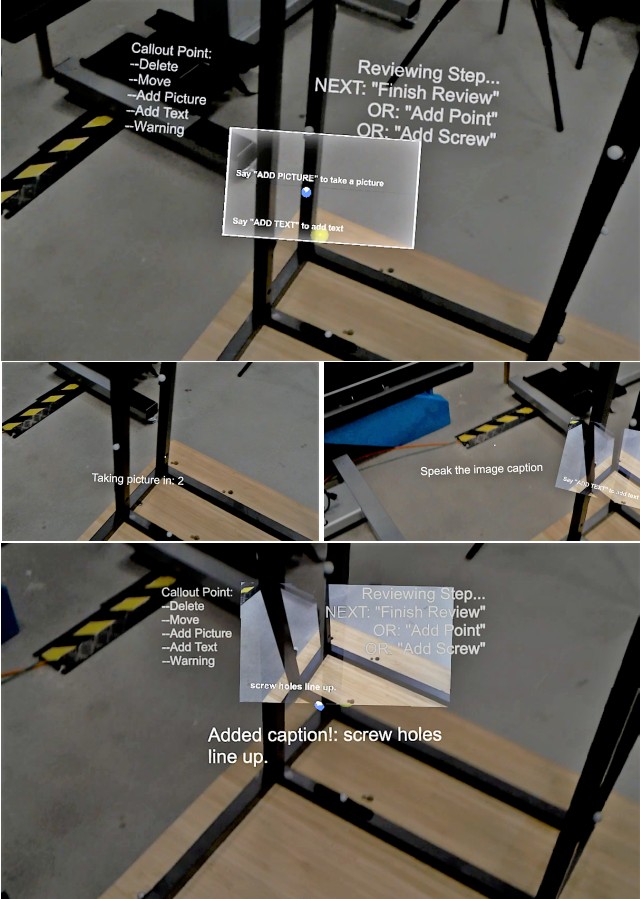

Figure 7: After adding a callout point, that point has a canvas to fill in (top). The author can add a picture (left), and a caption (right) and then has a completed callout point (bottom).

associates the spoken text as the image caption.

Authors can describe a callout point as a negative example or a warning simply by looking toward the callout point and saying "*Warning*". This defines that callout point as a warning or negative example and will draw tutorial users to be extra attentive when traversing the tutorial. The callout point turns red and the associated image is given a red border around it (Figure 8). Authors are also able to move the point along the surface of any tracked piece or delete it entirely.

Though fastening components together is an important aspect of an assembly tutorial, fastener objects (e.g., screws, nails) are too small for traditional object tracking technologies. During step recording, *AuthAR* records use of the tracked screwdriver and detects when it was used on a tracked material, generating a virtual screw hole at that location. Making use of these generated screw holes from step recording, the author can associate a virtual screw with the virtual screw hole by looking toward the hole and saying "*Add Screw*" (Figure 9). The user cycles through possible screws by saying "*Next*" and "*Previous*"-commands which cycle through possible virtual screw representations.

The author is able to hold the physical screw up to the virtual one for comparison and say "*This One*" to associate the screw with that hole. Authors can also manually add new fasteners in areas that cannot be tracked. To do so, the author once again says "*Add Screw*", pulling up the same menu, but when a screw has been selected, the ray-casted, gaze-based cursor allows the user to manually place the

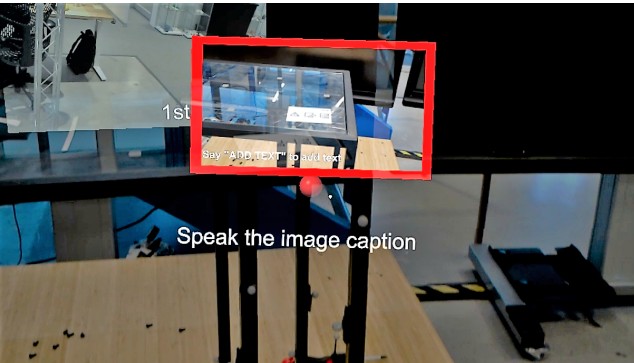

Figure 8: Tutorial author setting a warning about fragile glass using a red callout point and a red border.

virtual screw where it was physically placed. This is useful for the laptop stand, for example, as there are rubber feet that need to be screwed in by hand, rather than with the tracked screwdriver. Saying "*Finish Review*" brings the user back to the original idle state and the user has iterated through the step building process at this point.

## 6 DISCUSSION

Toward validating *AuthAR*, we discuss our initial observations in testing with tutorial authors, present an example application that parses and displays the generated tutorial for end users, and explain extensibility beyond the presented use case. In doing so, we consider improvements to *AuthAR*, and design considerations for other *in situ* AR content authoring tools.

### 6.1 Initial User Feedback

To gather initial observations and feedback of the system, we asked two users to generate a tutorial for an Ikea laptop stand. Though we have not formally evaluated *AuthAR* for usability, this initial feedback provides insight into possible improvements to *AuthAR* and generalized takeaways in building such systems. The standard paper instructions for the stand consist of four steps: fastening the legs to the bottom base, fastening the top to the legs, adding screw-on feet to the bottom of the structure and adding glass to the top. We guided the users through using the system while they created an AR tutorial for assembling this piece of furniture.

Both testers found the system helpful for generating tutorials with one noting that "being able to take pictures with the head for annotations was really useful." This suggests that the embodied gaze-based interaction is particularly well-suited to picture and video recording. Most of the functionality for making refinements in the tutorial is enabled by the user looking anywhere near the objects, however, adding new callout points requires accurate hovering of the cursor on the object of interest while speaking a command. One user mentioned that it was "kind awkward to point at certain points with the head". In such systems that require precise placement of virtual objects on physical components, pointing at and touching the position where the callout point would be a useful improvement over a gaze-only approach.

Though fulfilling the hands-free requirement of a tutorial generation system (**D1**), *AuthAR's* use of dictation recognition for text entry was particularly challenging for users, in part due to the automated prompting for step descriptions and titles. One participant was surprised by the immediate prompt for a step description, and said that "it was hard to formulate something articulate to say by the time it had finished recording", so future iterations will likely incorporate users explicitly starting dictation recognition for a title so they are prepared to give one.

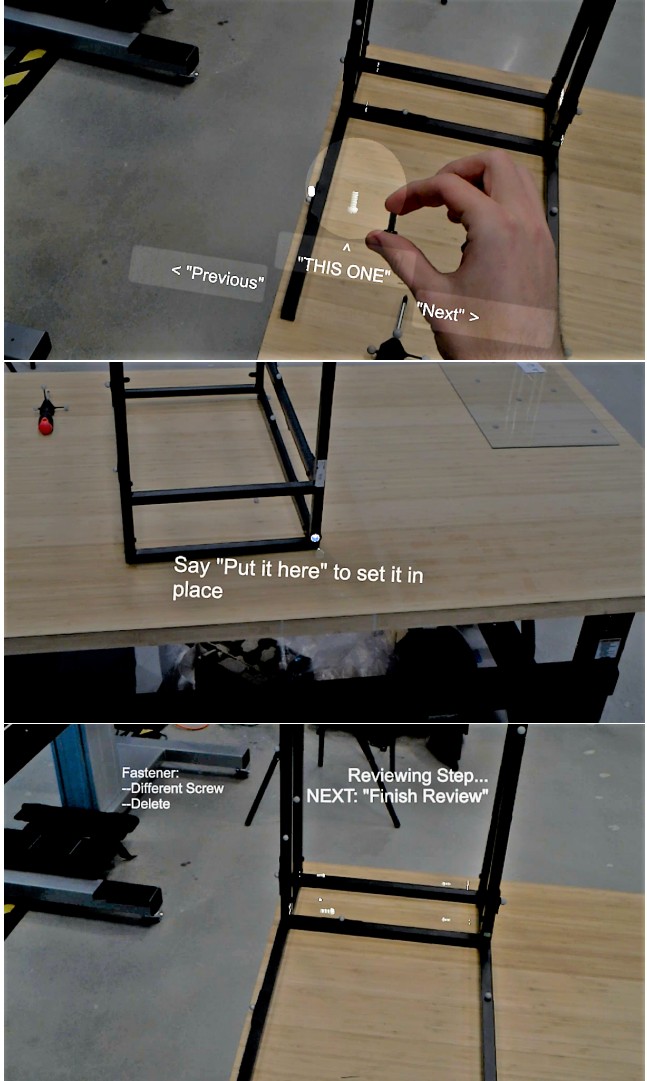

Figure 9: User adding virtual screws to the tutorial. The user can hold the physical screw up to the virtual one for comparison (top), and if the screw hole was not automatically generated, the user can place the screw via ray-cast from the headset (middle).

While we focus on authoring the tutorial in real time, the users wanted to view and refine previous steps. One possible enhancement to an *in situ* content authoring tool would be a "navigation prompt to let you jump between steps and review your work." With *AuthAR*, this would provide users the ability to review and refine the full tutorial at a high level, including previous steps.

### 6.2 Tutorial Playback/Walkthrough

Though we focus on authoring of assembly tutorials, we also implemented and tested a playback mode to validate *AuthAR's* ability to generate working tutorials. The tutorial author can save a tutorial after finishing a step, and *AuthAR* generates an XML representation of the entire tutorial. We load this XML into the playback application, also built for the HoloLens, and components of the steps built earlier are loaded and displayed for the end user to follow.

The guidance techniques are simple but demonstrate success of the authoring process and portability of the generated tutorial. Our playback application projects lines from each piece's location to

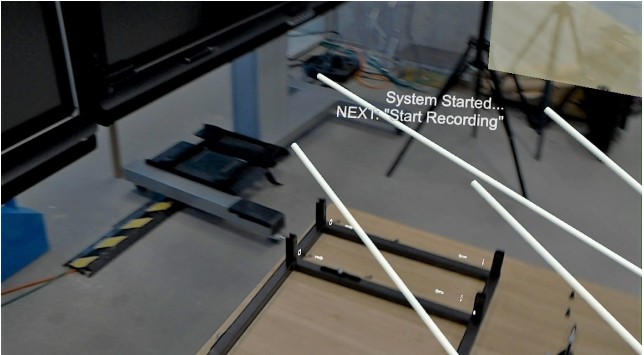

(a) Lines projected from piece location to goal location.

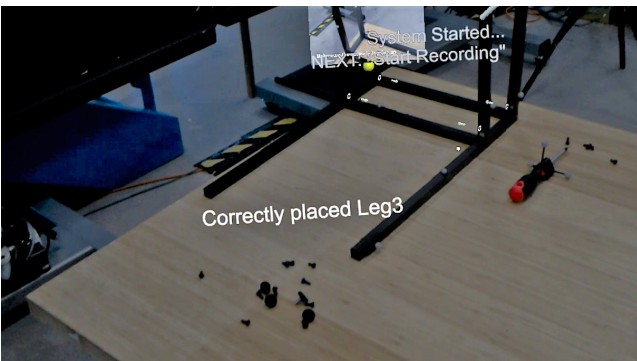

(b) Successful placement of pieces.

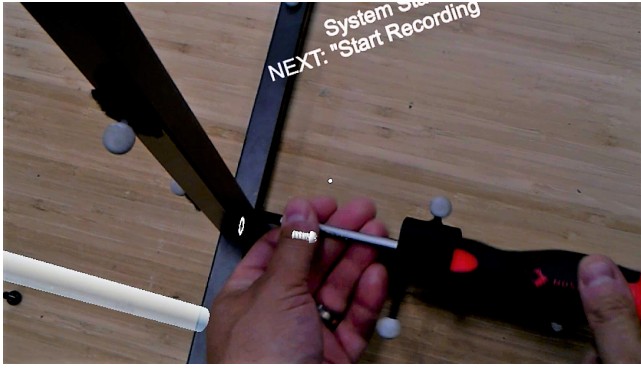

(c) Augmentations to guide screw placement.

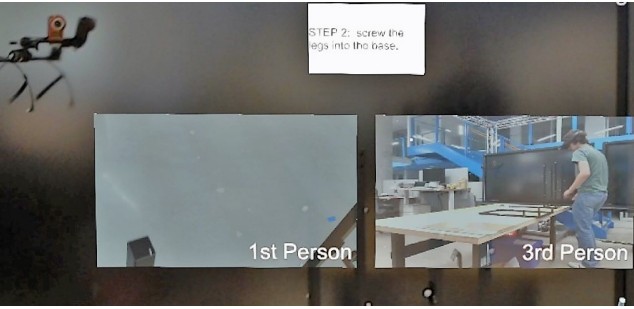

(d) Playback of first and third person perspectives.

Figure 10: Example playback of a generated tutorial.

where that piece needs to go (Figure 10a). When the end user of the tutorial has correctly placed the object, a notification appears in front of them (Figure 10b). The user also receives guidance of where add screws and when the playback application recognizes a screw event in that location, the screw augmentation disappears (Figure 10c).

Both first person and third person video representations play on a loop across the table from the user (Figure 10d). As the user walks around the table, the videos adjust their positions such that the user can always look up and see the videos straight across from them. Use of the third person video is currently the only tutorial component that requires post-processing after the *in situ* authoring process is complete. Because the XML representation of the tutorial uses file paths for videos, the author needs to manually move the video from the Android tablet to the headset's file system. Future iterations could automatically stream these videos between components.

### 6.3 *AuthAR* Extensibility

We demonstrate *AuthAR* with assembly of an Ikea laptop stand but note that it could be extended to any physical task where simplified virtual models of the pieces could be built or obtained. All of the pieces used for the laptop stand were scaled cubes or combinations of scaled cubes, with disabled "Renderer" components to create the illusion of adding virtual augmentations to physical parts, when in reality, there were invisible objects overlaid on top of the physical pieces. So simply loading in pieces and disabling the visual renderings allow for extensibility to virtually any assembly task.

While demonstrated with an augmented screwdriver, *AuthAR* could be extended to support different tools, or an integrated workspace of smart tools [23, 37]. We employed very simple logic, that whenever the tip of the screwdriver hovers near a piece, *AuthAR* adds a screw hole. Future iterations could employ more advanced logic for detecting tool usage. The need for a large truss with 10 Optitrack cameras enables very accurate tracking of visual markers but limits the *AuthAR's* widespread deployment in its current state. For practical use we envision the localization of objects being done with the headset only, or with cheaper external optical tracking, perhaps with black and white fiducial markers. In this scenario, tracked pieces could be established in real time though user-assisted object recognition [24], rather than defining their shapes prior to running *AuthAR*.

The *in situ* authoring approach offered by *AuthAR* allows the user to concurrently craft the tutorial while assembling the pieces. However, the gaze/voice multimodal interface does not provide users with efficient tools to fine-tune the generated tutorial. To this end, *AuthAR* should be complimented by a 2D interface for precise editing of individual components of the tutorial, similar to tutorial generation tools described in previous work [6, 7, 16]. Trimming 1st and 3rd person perspective videos, cropping images and editing text are currently not well-supported by *AuthAR* and would be better suited to a mouse and keyboard interface. This complimentary approach would also allow users to focus on coarse-grain tutorial generation and object assembly without being burdened by these smaller edits that can easily be done after the fact.

### 7 CONCLUSION

*AuthAR* enables tutorial authors to generate mixed media tutorials semi-automatically to guide end users through the assembly process. We automatically record expert demonstration where possible and allow for *in situ* editing for refinements and additions. We built *AuthAR* with several design guidelines in mind, validated with the authoring of a tutorial for assembling a laptop stand, and discuss the extensibility to assembly of other tasks by simply loading different virtual models into *AuthAR*. We see *AuthAR* enabling authoring of tutorials that could reach a widespread population with mixed media tutorials flexible to the preferences of each individual user.

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
