# OpenReview forum: "AuthAR: Concurrent Authoring of Tutorials for AR Assembly Guidance"
_graphicsinterface.org/Graphics_Interface/2020/Conference — GI 2020_

### Official Review · AnonReviewer2 · 2020-01-08
**The paper introduces a system for authoring AR-based tutorials for the assembly of physical objects. The design of the system is currently based on preliminary empirical work. Yet, the proposed solution deals with several design challenges, and therefore, I would recommend acceptance**

**Confidence:** 4
**Rating:** 7

**Review:**

The paper presents a system for authoring augmented-reality (AR) tutorials for manual construction tasks, such as assembling IKEA furniture. The system combines multiple instruction representations, in particular, video, images and text. It allows tutorial authors to record and then review their physical interactions from different perspectives (first view + third view). The paper does not present any informative user study or formal evaluation but discusses results of a preliminary evaluation with two participants (acting as authors). The submission further includes a video that demonstrates key features of the system.

The work is still in progress, and the system definitely requires further design iterations before being usable. On the other hand, building a system that combines tracking from multiple sources (motion tracking system + tablet + HoloLens) is challenging  and can be considered as a contribution on its own. The presented scenario provides insights about the technical but also interaction-design challenges for building such systems and their limitations. Therefore, I would recommend (weak) acceptance. That said, I also believe that this work is not complete yet, and the paper can be significantly improved.

These are some major limitations of the current work and some suggestions for thought and improvement:

- Although I understand the the focus of the paper are the authoring tools, it is hard to think about the design of such a system without carefully considering the quality of the produced tutorials from the perspective of a novice maker who needs to assembly an object. The paper provides very little discussion about the needs of users who follow a tutorial and how the system tries to meet them.

- The lack of a formal evaluation is definitely an important weakness of the current work. The informal study indicates several limitations and I wonder how easy it is to use such as a system, in particular when tracking fails or is not well precise.

- I am further concerned about the quality of the final tutorials. How easy it is to create a comprehensive tutorial, which is at least better than a simple video showing a person assembling the object? I agree that providing multiple representations is valuable, but at the same time, this complicates the authoring process. Unfortunately, the paper does not provide clear insights about effective techniques of combining different representations. Are there any standards or recommendations about how to create successful tutorials or individual assembly instructions? I encourage the authors to check related documentation or research work and come with some specific (well argued) guidelines. If there is no related work (I am not very familiar with this topic), I think that it is worth running a design study with both experts and novice assemblers.

- The paper presents some interesting ideas about how to guide users to choose and drive screws. It is further inspired by some common techniques, as the ones shown in Figure 6, on how to draw a user's attention. I encourage the authors to look into such techniques in more detail and try to identify some more generic principles for their tutorial designs. Similarly, the authors may need to identify concrete sub-tasks in such assembly scenarios and their challenges (e.g., choosing the correct screw, fitting a piece in the correct direction) and then propose alternative techniques to guide them.

- The paper claims that generalizing the approach could be extended to any physical task. However, it seems to me that the models of individual tracked pieces is preconfigured, and the current system does not provide any tools for facilitating this process. Furthermore, tracking can become problematic in several scenarios, in particular when parts are visually occluded. I think that the paper would need to provide further evidence to make such claims.

- The first sections of the paper are very well written. However, the later ones that describe the actual system are not easy to follow. It would be probably better to start with a walkthrough scenario to explain the use of the system. Then, the paper could summarize key interaction principles, justify their design, and further emphasize their contributions.

---

### Official Review · AnonReviewer5 · 2020-01-08
**The authors present a system that facilitates authoring of augmented reality tutorials.**

**Confidence:** 3
**Rating:** 4

**Review:**

The paper is well written and easy to follow. The system seems to be well implemented and the video is well made. However, it is not clear, what the research contribution of this work is. In the following, I would like to elaborate on the main issues of this work.

-- Technical contribution

Even though well implemented, the system is based on simple components and a simple overall architecture. Therefore, there is no contribution from a technical standpoint.

-- Design contribution

The design of the system and the interactions are straightforward (e.g., based on standard HoloLens interaction techniques). The authors claim the following: "Our design of the AuthAR system was grounded by our exploration of the assembly task design space". It is unclear, in which way the authors "explored" the assembly task design space. All design considerations are fairly obvious and not based on, e.g., domain experts in assembly lines or a thorough identification of challenges in the authoring process.

-- Scope

The topic that the authors chose is relevant and the challenges that the authors identified make sense from a generic high-level perspective. However, the authors did not clearly define a scope and the paper and ideas remain on a superficial level.
Primarily, the authors do not clearly define a target group. Is the goal to enable tutorial authors without video editing knowledge to create tutorials, i.e., to allow a broader audience to generate tutorials? Or is it about reducing cost? The introduction seems to imply the latter. More importantly, is the system intended to be used for anything from furniture assembly (like, e.g., hinted in the video) to more professional assembly lines (PCB assembly, assembly of machinery etc.)? Even though those are all assembly tasks superficially speaking, their requirements, authors, end-users etc. vastly differ. The authors need to specify, for which scenarios their system meets the requirements. For instance, if a furniture assembly tutorial is authored so that end-customers can be guided through the assembly at home, then this tutorial will be used by thousands or even millions of customers. This means that, making a properly produced tutorial seems a lot more valuable than saving comparatively small costs in the creation process. On the other hand, for professional assembly lines, the tasks might be more specialized, subject to regular change and tutorials will only be used by few employees. In this case, a cost-saving quick creation process of tutorials like proposed by the authors might be very beneficial. But with that, some other, potentially interesting challenges arise. For instance, what if a step needs to be changed? Furthermore, products become increasingly personalized with an increased demand for customization. A system like the one proposed by the authors could potentially allow for authoring dynamic tutorials that play back the correct steps depending on the variant of the product. As of now, the paper entirely lacks reflection and a clear definition of goals.

-- Dependency on predefined 3D content

The currently supported co-located augmentations are limited to skrews. The authors mention that representations for other parts and tools can be implemented in the future. However, I think the problem is a bit deeper than that. While it is not a requirement for a research prototype to support a lot of different types of content, it is questionable whether the overall principle meets the requirements of low-cost and easy creation of AR tutorials. The reason is that assembly steps often require specific 3D content that represent parts and specific procedures for attaching them. All of those would need to be programmed and then accessed by the author, similar to the skrews and the tracked skrewdriver that served as an example in the paper. Technically, the authoring process already starts with generating and programming such virtual representations. Recording the actual steps in the end might be low-cost, but generating the needed 3D content beforehand and selecting them from a large database of virtual parts and pre-programmed behavior increase the cost heavily. To meet the requirements and claims of the authors, it might therefore be preferable to devise a solution that does not rely on predefined specific 3D content. For this matter (and also more generally), the authors might want to seek inspiration from remote collaboration or telepresence research systems, that do not rely on predefined 3D content, but instead utilize 3D reconstructions.
Example: Gao et al. 2016 "An oriented point-cloud view for MR remote collaboration".
Analogously, an authoring system could use 3D video capture to fully generate co-located instructions.


In summary, due to the very basic design, implementation and evaluation I cannot recommend acceptance of this work. The clarity and good presentation increases the value of this work, but due to the lack of a research contribution, I think that publication cannot be warranted at this point. For future iterations, I suggest that the authors define a clear scope and identify challenges a lot more thoroughly.


Lastly, here are some minor issues:
-The "design guidelines" section header should be renamed, since the design decisions are based on observations and not guidelines. Maybe "design rationale" ?
-The authors occasionally jump to implementation details and terminology (e.g., "renderer component"). Furthermore, the descriptions are oddly phrased (e.g., using an "invisible object with the same shape as the physical object"). At the same time, those details are not important for reproducibility and can be removed.
-Playback is only described in a sub section towards the end. However, I consider this to be a rather crucial part of the overall system that should be discussed earlier and in conjunction with the authoring process.
-The "Discussion" section is not structured well. The sub sections (validation, playback, future work) have a very weak connection. One possible fix might be to remove the "Discussion" header and paragraph, make the user feedback its own section, move playback to earlier parts of the paper and create a dedicated "Limitations and Future Work" section (in connection to this, a discussion about limitations is currently lacking as well).

---

### Official Review · AnonReviewer1 · 2020-01-10
**Overall good paper**

**Confidence:** 3
**Rating:** 7

**Review:**

The paper describes the technical details of the system well. But, it can be further improved by talking about how the design guidelines and design space are derived and improving it’s validation framework.
This paper presents a design for an augmented reality (AR) based authoring tool for assembly tutorials.

It proposes a system which attempts to generate tutorials with mixed media, and also allows the author to generate the tutorial while performing the assembly in-situ requiring minimal to no post-production. Following presenting an overview of the current literature in AR based tutorials and authoring, a set of design guidelines and the relevant design space is described. The use of these elements to guide the design of the system is a good justification made by the authors of the paper. Following which the system design and the tutorial authoring process is described. Finally, the authors discuss some initial user feedback and possible ways in which the system can be extended.

While the approach to the problem is interesting, some of the concerns I had with the overall system and the study following are as follows:
• How the guidelines and the design space are derived is not clear; this isn’t discussed in any of the previous work in any form? or is it based on some empirical evidence?
• How does these guidelines and design space described compare to the design choices made in previous work?
• How is the design choices for the heads-up-display (HUD) made, are they grounded on any other previous work? and how does it influence the overall experience of authoring a tutorial?
• The validation of the system is weak. It is unclear as to what is implied here by "validation", is it simply assessing that the system works, is it being compared to another system or it being validated using a study? How successfully does the system address the limitations of prior work? how does it compare to the previous systems? What were the approaches taken by authors of the previous systems to validate their system? are any of them applicable here? if not, why? How is the generated tutorial compare to a tutorial generated by other systems or done manually?
• In the user feedback, the background of the users is not provided. Do they have any experience in authoring tutorials? do they have any related expertise to the assembly process? Did they have anything to compare their experience against? Since colours used in the interface, has the design taken into consideration colour blindness or were the participants tested for colour blindness. Ideally for a system of this nature, it would require two sets of validations, one by the authors of a tutorial, another by end-users who use the tutorial, since there a generated components in the tutorial. Alternatively, if the proposed system is going to have a middle phase to generate the tutorial for the end-user which is not in the scope
of this paper, it is not made clear.
• The paper can benefit if it had a separate section for limitations and future work, rather than bundling it into the discussion. Also the "Tutorial playback/walk-through" sub-section seems like a better fit for the section "The AuthAR System" as what the output is and how it is generated is also a component of the system.

Other general fixes in the paper:
• On page 5, under the "Interaction Paradigm" sub-section, there is a paragraph that is not formatted correctly.
• The citations are inconsistent in formatting. There are multiple citations that only have the authors and a title, what are they? Also what is the citation number 48?

---

### Meta-Review · Area_Chair1 · 2020-01-10

**Recommendation:** Accept
**Confidence:** 4

**Metareview:**

2 out of 3 reviewers recommend (weak) acceptance of this work, as such I will stick with the majority. Reviewers found the paper to be well written and that the solution dealt with several design challenges.  Also, all the reviewers also think that there are major limitations and a lack of focus and provide detailed feedback on how to improve. I hope the authors will find the insightful and constructive criticism provided by the reviewers beneficial for their future works.

---

### Decision · Program_Chairs · 2020-01-11

Accept